# Geochemical Characteristics and Controlling Factors of Chemical Composition of Groundwater in a Part of the Nanchang Section of Ganfu Plain

**Qingshan Ma \*, Weiya Ge \* and Fujin Tian**

Nanjing Center, China Geological Survey, Nanjing 210016, China; tianfj@mail.cgs.gov.cn
\* Correspondence: maqingshan@mail.cgs.gov.cn (Q.M.); gewy@mail.cgs.gov.cn (W.G.)

**Abstract:** This work aims to investigate the hydrochemical characteristics and formation mechanisms of shallow groundwater in a part of the Nanchang section of Ganfu plain. The hydrochemical data from 90 groundwater samples were interpreted by the methods of mathematical statistics, Piper diagrams, Gibbs plots, ratio graphs of ions, and geochemical modeling. The results show that shallow groundwater is weakly acidic, the average concentration of cation in groundwater decrease in $Ca^{2+} > Na^+ > Mg^{2+} > K^+$, and the abundance is in the order $HCO_3^- > NO_3^- > SO_4^{2-} > Cl^-$ for anions. The hydrochemical type of groundwater was dominated by $HCO_3$-Na·Ca·Mg, $HCO_3$·Cl-Na·Ca·Mg, and $HCO_3$-Na·Ca. Moreover, the main controlling factor of groundwater hydrochemistry is water-rock interactions. $Na^+$ and $K^+$ mainly originate from the dissolution of halite. $Ca^{2+}$ and $Mg^{2+}$ are mainly controlled by carbonate dissolution, while the main anions come from the dissolution of evaporite and carbonate. The groundwater chemical evolution is affected by the dissolution and precipitation of the mineral phase and cation exchange.

**Keywords:** shallow groundwater; hydrogeochemical processes; geochemical modeling; Nanchang

## 1. Introduction

Groundwater is a valuable natural resource and plays a vital role in social production, life, and ecosystem maintenance, especially in arid and semiarid regions. Due to the frequent occurrence of extreme climate and the rapid growth of population, the demand for groundwater resources has increased greatly in recent decades [1]. In addition to considering groundwater quantity, groundwater quality is also an essential factor in determining the availability of groundwater resources [2]. Generally, the hydrochemical composition of groundwater depends on the natural factors such as recharge water characteristics, aquifer media, and water-rock processes [3]. However, with the deepening of the impact of human society on the geological environment, groundwater is increasingly affected by mining, industry, agriculture, urbanization, and other human activities. Therefore, to analyze the chemical characteristics and formation mechanism of groundwater, especially in areas affected strongly by human activities, is the key to the safe and sustainable utilization of groundwater resources.

In recent times, a great deal of work has been done in assessing and protecting groundwater quality by many scholars all around the world. As an important research method, hydrogeochemistry specializes in the formation of chemical components and the migration, transformation, and enrichment of various elements in groundwater. It has been widely used to evaluate the chemical characteristics of groundwater [4]. Li et al. (2010) used geochemical modeling to quantify the evolution process and formation mechanism of local groundwater chemistry in the southern plain of Pengyang County, Ningxia, China [5]. Sivakarun et al. [6] adopted the methods of ion ratios and hydrochemical parameters statistics to illustrate the factors that affected groundwater in coastal areas of southern India [6]. Osta et al. [7] revealed the hydrochemical characteristics and the main evolution

processes of groundwater in sandstone aquifer. Liu et al. [8] analyzed the hydrochemical characteristics and the controlling factors of groundwater in the alluvial-proluvial fan of the Qinhe river using these methods of mathematical statistics: the Schakerev classification, Piper diagrams, Schoeller diagrams, Gibbs plots, and ion ratios.

Nanchang is situated in the north-central part of the Jiangxi Province and the alluvial plain of the lower reaches of Gan River and Fu River. It is the only capital city that is adjacent to the Yangtze River Delta, the Pearl River Delta, and the Western Taiwan Straits Economic Circle. As an important source of water, groundwater has a close connection to the development of Nanchang city. In recent decades, industrialization, urbanization, and economic growth have significantly impacted the groundwater environment. Based on the analysis of hydrogeological conditions in the study area, this work adopted the methods of mathematical statistics, Piper diagrams, Gibbs diagrams, ion ratios, and a hydrogeochemical model to reveal the geochemical characteristics and controlling factors of the chemical composition of groundwater in a part of the Nanchang section of the Ganfu plain. This work is helpful to understand the groundwater circulation process and provides a scientific basis for the evaluation and rational development of regional water resources.

## 2. Materials and Methods

### 2.1. Study Area

The study area is a part of the Ganfu plain and lies between 28°21′2.93″ and 28°46′15.13″ N latitude and 115°48′43.09″ and 116°07′32.06″ E longitude (Figure 1). It is located in the humid subtropical climate zone with a long humid summer and a short and cool winter. The annual average temperature is 17~17.7 °C. The average annual rainfall is approximately 1610 mm, with 49.33% concentrated in April through June. The average yearly evaporation is about 1227.4 mm [9].

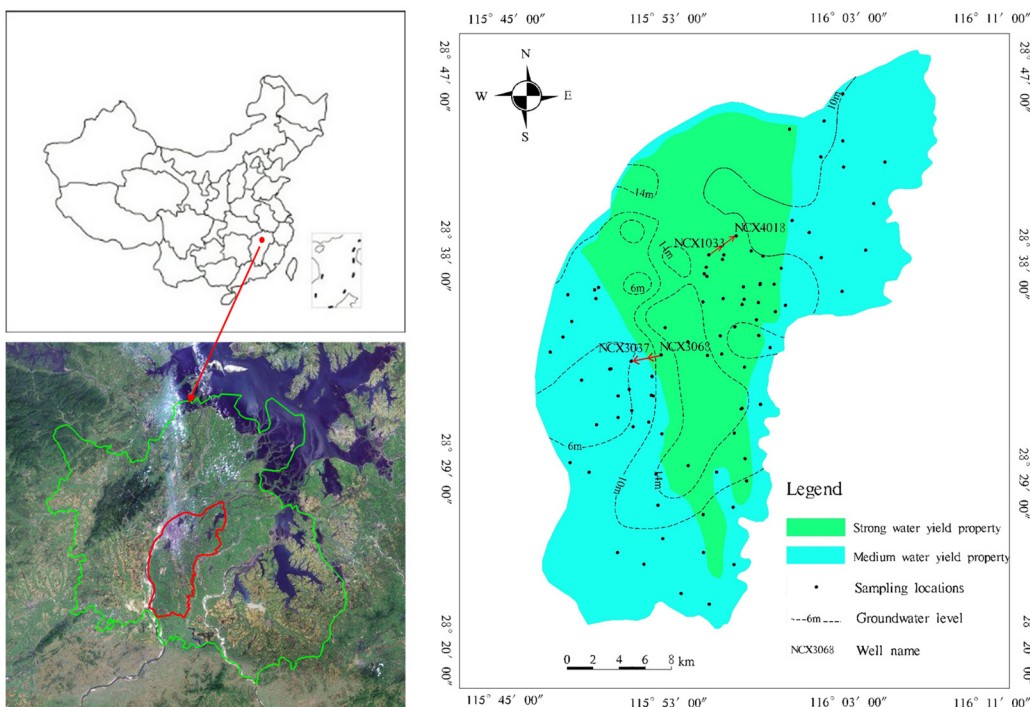

**Figure 1.** Location and sampling site map of the study area.

The groundwater in the study area is mainly quaternary loose rock pore water. The quaternary aquifers consist of sand and gravel aquifers of the upper Middle Pleistocene, Upper Pleistocene, and Holocene, respectively. The aquifer has a binary structure: the upper part is a relatively impervious layer, which is composed of cohesive soil and silt, with mixed mucky cohesive soil locally, and the thickness is generally 5~15 m, and the

average thickness is 6.95 m. The lithology of the lower part is sand and gravel, which is the main storage space for groundwater.

The bedrock fissure aquifer is widely distributed below the quaternary loose pore layer and has a good connectivity with the overlying loose rock pore aquifer [9]. The hydrogeological profile is shown in Figure 2.

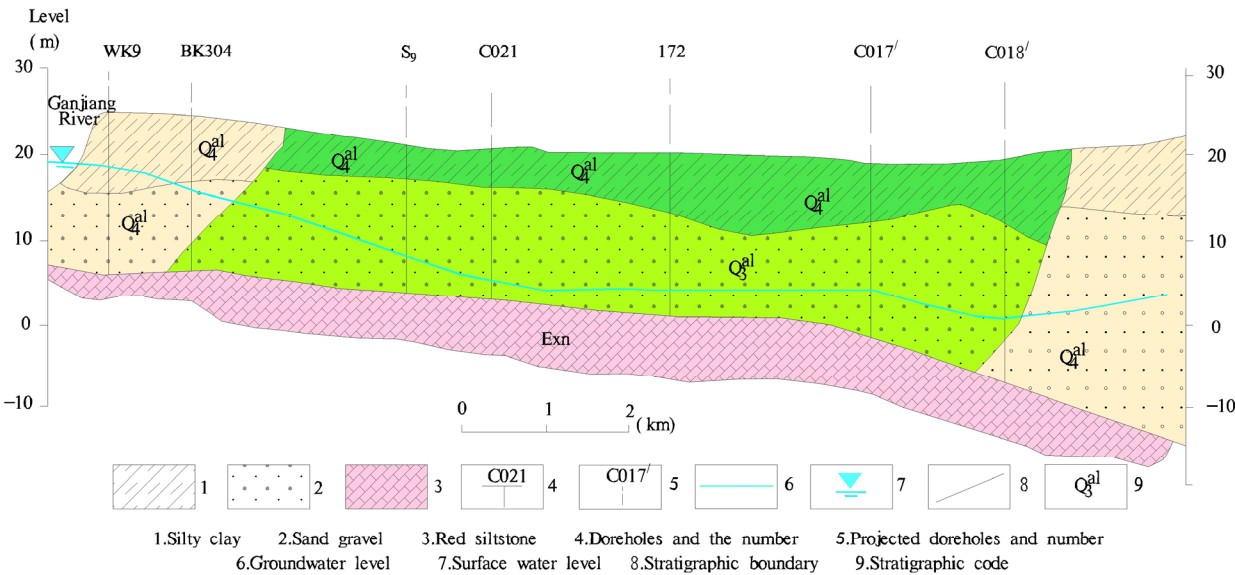

**Figure 2.** Hydrogeological profile of the study area from Bayi Bridge to Nanchang Iron and Steel Plant.

## 2.2. Sample Collection and Chemical Analysis of Groundwater

A total of 90 groundwater samples were collected in two sampling campaigns (December 2018 and July 2019). Groundwater samples were collected from shallow wells ranging in depth from 2 to 30 m below ground level. The clean, dry polyethylene plastic bottles were used to collect water samples, and the bottles were rinsed at least three times with the source water before sampling. During sampling, water was filtered by 0.45 μm pore size membrane and aliquots for cations acidified to prevent mineral precipitation. After sampling, the groundwater samples were labeled, sealed, and transported to the laboratory for testing within 48 h.

The parameters of temperature, DO, EC, Eh, and pH were determined by SMART SpectroTM on-site, and the others were tested by the Ministry of Land and Resources East China Mineral Resources Supervision and Testing Center. Flame atomic absorption spectroscopy (AAS) was used to determine the concentrations of four different metal cations ($Ca^{2+}$, $Mg^{2+}$, $Na^+$ and $K^+$) (Perkin Elmer AAnalyst 100). The total hardness and $HCO_3^-$ were titrated using a HACH digital titrator. $Cl^-$, $SO_4^{2-}$, $NO_3^-$, $F^-$ were measured on ion chromatography (Dionex ICS-2500), and the total dissolved solids (TDS) were determined by the drying method [10]. The reliability of chemical data was checked for accuracy via ionic balance analysis. The results show that the ionic balance error is less than ±5%, indicating that the accuracy of the measurements is acceptable in this study [11].

## 2.3. Analytical Method

Combined with the hydrogeological condition and hydrochemical data of the study area, the chemical characteristics of groundwater were analyzed using the methods of mathematical statistics, Piper's trilinear diagram, and Gibbs' diagram, etc. [12,13]. The hydrogechemical calculation (saturation indices, logPCO2) and modeling (inverse modeling) were carried out using PHREEQC 2.8 software [14]. A Piper diagram was used to analyze the evolution characteristics of groundwater hydrochemistry. A Gibbs diagram was used to analyze the formation mechanism of chemical component of groundwater. The reverse hydrogeochemical simulation was used to determine the amount of dissolution

or precipitation of minerals and to analyze the variation characteristics of each hydrochemical component along the flow path. Descriptive statistical analysis of hydrochemical parameters was carried out by SPSS 22.0 software. Piper's trilinear diagram and Gibbs' diagram are drawn by AqQA 1.1.1 [15] and Excel software, respectively. Other diagrams were drawn using Photoshop and Arcgis 10.0.

## 3. Results and Discussions

### 3.1. Statistical Results

The statistical results of the analyzed parameters have been presented in Table 1. Generally, the groundwater is weakly acidic with mean acidity being 6.27 pH units. The pH data display little variability over the domain of the study area. The TDS values of groundwater range between 38 mg/L and 593 mg/L with an average of 198.46 mg/L. $Na^+$ and $Ca^{2+}$ are the predominant cations. The concentration of $Ca^{2+}$ varies from 1.5 to 105 mg/L with a mean of 26.4 mg/L. The concentration of $Na^+$ varies from 4.01 to 85.2 mg/L with a mean of 19.79 mg/L. The average concentration of cation in groundwater decreases in $Ca^{2+} > Na^+ > Mg^{2+} > K^+$. Calcium ($Ca^{2+}$) is the most abundant cation in groundwater, but most samples are obviously much less than the WHO standard (75 mg/L) for drinking water (WHO, 2006) [16]. With respect to anions, $HCO_3^-$ is the most dominant anion, and the concentration ranged between 5.51 and 352 mg/L. The abundance is in the order $HCO_3^- > NO_3^- > SO_4^{2-} > Cl^-$ for anions. The dominance of the $HCO_3^-$ ion also suggests that groundwater in the area is generally shallow groundwater [17].

**Table 1.** Statistical results of chemical parameters of water samples.

| Parameters | Maximum | Minimum | Average | Standard Deviation | Coefficient of Variation |
|:---:|:---:|:---:|:---:|:---:|:---:|
| pH | 7.62 | 5.02 | 6.27 | 0.51 | 0.08 |
| $K^+$ | 79.00 | 1.10 | 8.99 | 12.31 | 1.37 |
| $Na^+$ | 85.20 | 4.01 | 19.79 | 12.72 | 0.64 |
| $Ca^{2+}$ | 105.00 | 1.50 | 26.40 | 18.33 | 0.69 |
| $Mg^{2+}$ | 28.80 | 0.80 | 10.85 | 5.70 | 0.53 |
| $Cl^-$ | 71.40 | 5.64 | 22.53 | 14.16 | 0.63 |
| $SO_4^{2-}$ | 162.00 | 0.04 | 33.60 | 26.96 | 0.80 |
| $HCO_3^-$ | 352.00 | 5.51 | 83.96 | 65.16 | 0.78 |
| $NO_3^-$ | 144.56 | 0.02 | 34.38 | 29.73 | 0.86 |
| TH | 362.00 | 7.00 | 110.36 | 64.40 | 0.58 |
| TDS | 593.00 | 38.00 | 198.46 | 110.80 | 0.56 |
| EC | 711.67 | 72.45 | 269.29 | 133.96 | 0.50 |
| Eh | 239.76 | −2.10 | 92.35 | 55.10 | 0.59 |

Values are in milligrams/liter unless otherwise stated.

The coefficient of variation shows that there is more pronounced variability in the spatial distribution of ion concentrations. The spatial distribution of main ion concentrations is shown in Figure 3. It can be seen from Figure 3 that the concentration of ions has an apparent zoning phenomenon, which generally presents the distribution characteristics with high concentrations in the central part of the study area and low concentrations in the north and south sides.

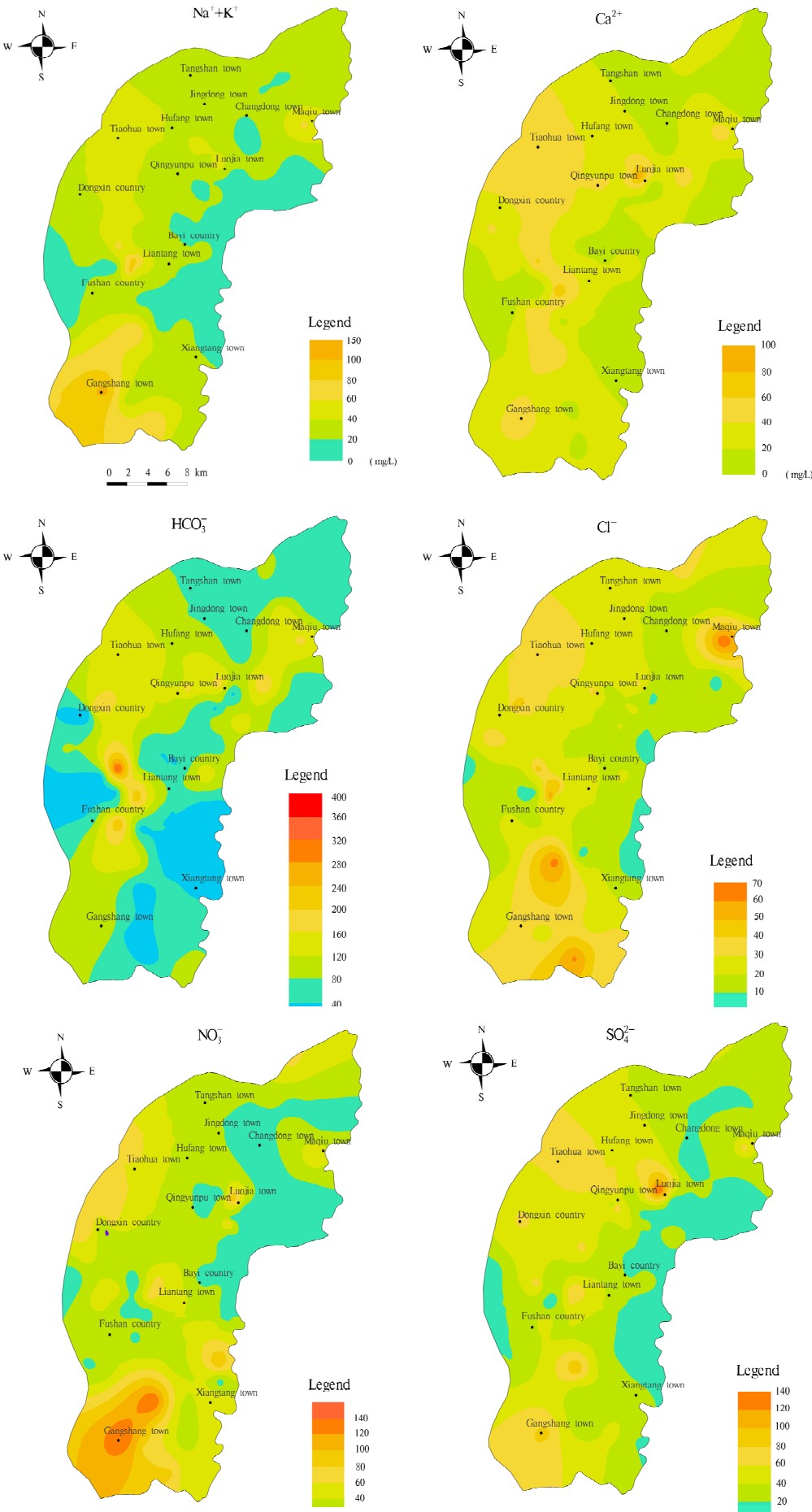

**Figure 3.** Spatial distribution of main ion concentrations.

### 3.2. Hydrochemical Types

Piper diagrams have been widely used to characterize the composition and evolution of the main ions in groundwater chemistry. The method has an advantage of being independent of human influence [18]. The groundwater samples were put on the Piper diagram, as shown in Figure 4. On the cationic triangle (left-hand triangle), most of groundwater samples were classified as calcium type or mixed type, collectively accounting for 73.34% of water samples. On the anion triangle (right-hand triangle), the bicarbonate and mixed types were predominant, collectively representing 82.71% of all groundwater samples. As seen on the diamond of the Piper plot, the hydrochemical types of groundwater in the study area belong to three categories, including the $HCO_3$-Na·Ca·Mg facies, $HCO_3$·Cl-Na·Ca·Mg facies, and $HCO_3$-Na·Ca facies.

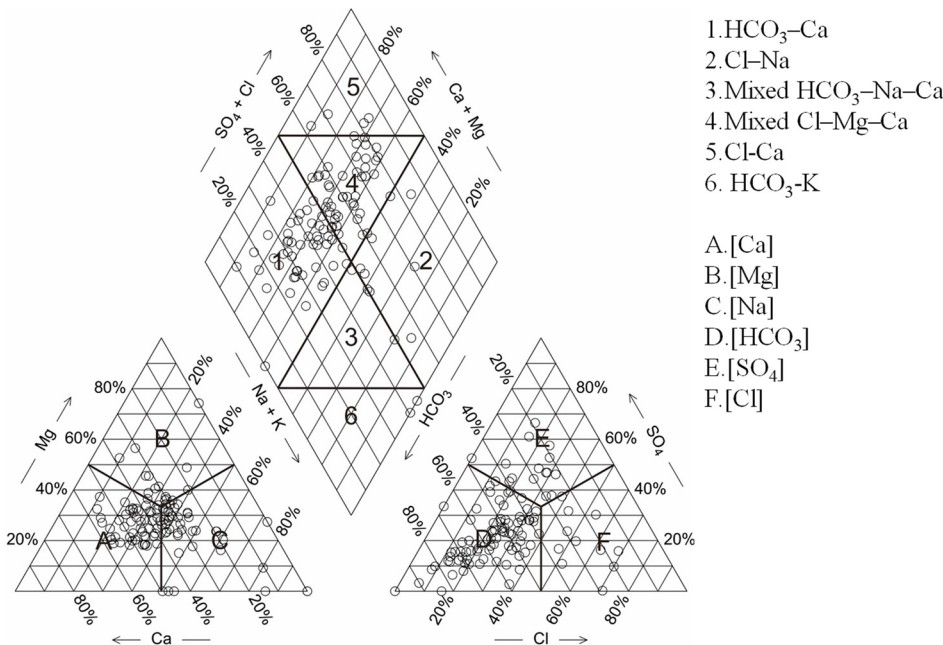

**Figure 4.** Piper diagram.

### 3.3. Mechanisms Controlling Groundwater Geochemistry

#### 3.3.1. Gibbs Diagram

Gibbs diagrams can be utilized to qualitatively determine the hydrochemical evolution process controlled by atmospheric precipitation, evaporation, and rock weathering, which were originally drawn using surface water samples [13]. If Gibbs diagrams are used to analyze groundwater, the outlines of Gibbs diagrams should be reconsidered [19]. As shown in Figure 5, the ratio of $Na^+/(Na^+ + Ca^{2+})$ ranged from 0.2 to 0.89, whereas the $Cl^-/(Cl^- + HCO_3^-)$ anion weight ratios ranged from 0.05 to 0.72. From the distribution of groundwater samples, most of the collected samples were located in the central area of the Gibbs diagram, indicating that the chemical composition of groundwater in the study area was mainly dominated by water–rock interaction. Besides, the increase in the ratio of $Na^+/(Na^+ + Ca^{2+})$ implied that cation exchange also impacts groundwater hydrochemistry [20].

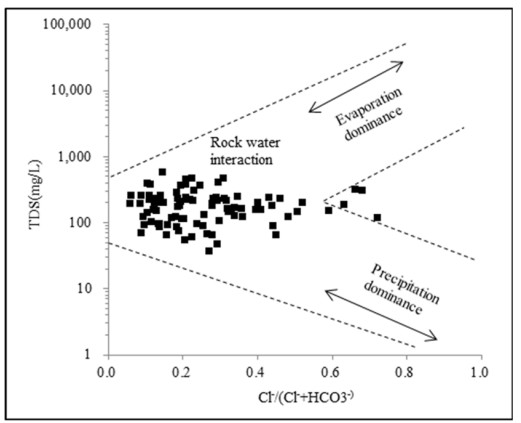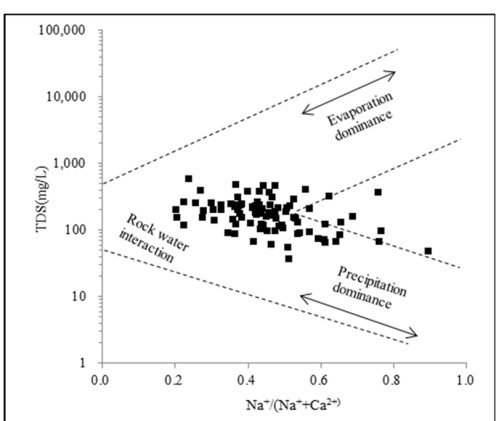

**Figure 5.** Gibbs map of groundwater in the study area.

3.3.2. Hydrogeochemical Processes

To further analyze the source of chemical components of shallow groundwater, the ratio graphs of ions are used to determine the origins of solutes and hydrochemical processes [21].

The ratio of $Na^+ + K^+/Cl^-$ has been widely used to study the source of $Na^+$ and $K^+$. If the ratio of $(Na^+ + K^+)/Cl^-$ is greater than 1, it indicates that hydrochemistry is mainly influenced by the dissolution of halite. Otherwise, the dissolution of silicate drives the groundwater hydrochemistry of the region. Figure 6a shows that almost all samples laid above the line ($Na^+ + K^+$: $Cl^-$ ratios varied from 0.86 to 3.59), indicating that $Na^+$ and $K^+$ originate mainly from the dissolution of halite.

A ratio of $(Ca^{2+} + Mg^{2+})/(HCO_3^- + SO_4^{2-})$ can be used to reflect the dissolution of carbonate and sulfate in groundwater. If the ratio of $(Ca^{2+} + Mg^{2+})/(HCO_3^- + SO_4^{2-})$ is greater than 1, it indicates that $Ca^{2+}$ and $Mg^{2+}$ originate from the dissolution of carbonate. Otherwise, it means that $Ca^{2+}$ and $Mg^{2+}$ originate from the dissolution of silicate and evaporite. As it can be seen in Figure 6b, approximately 57.77% of the samples laid above the 1:1 line, indicating that the dissolution of carbonate was the predominant processes of groundwater. Meanwhile, the dissolution of silicate and evaporite also occurred.

A ratio of $(SO_4^{2-} + Cl^-)/(HCO_3^-)$ could be used to determine the main source of $SO_4^{2-}$ and $Cl^-$. If the ratio of $(SO_4^{2-} + Cl^-)/(HCO_3^-)$ is greater than 1, it indicates that the chemical composition of groundwater is influenced by evaporite. Conversely, that is affected by carbonate. As shown in Figure 6c, most of the water samples were on both sides of the 1:1 line, and almost 54% samples laid above the line, and 46% samples laid below the 1:1 line, indicating that the dissolution of carbonate and evaporite were the predominant processes of groundwater in the region.

The ratio of $Ca^{2+}/Mg^{2+}$ has been widely used to study the dissolution of major minerals [22]. The relationship between $Ca^{2+}$ and $Mg^{2+}$ is shown in Figure 6d. Most samples laid above the 1:2 relationship line, indicating that groundwater hydrochemistry was affected by the dissolution of silicate. However, approximately 24.4% of the samples were located between the 1:1 and the 1:2 line, suggesting that the dissolution of calcite also occurred.

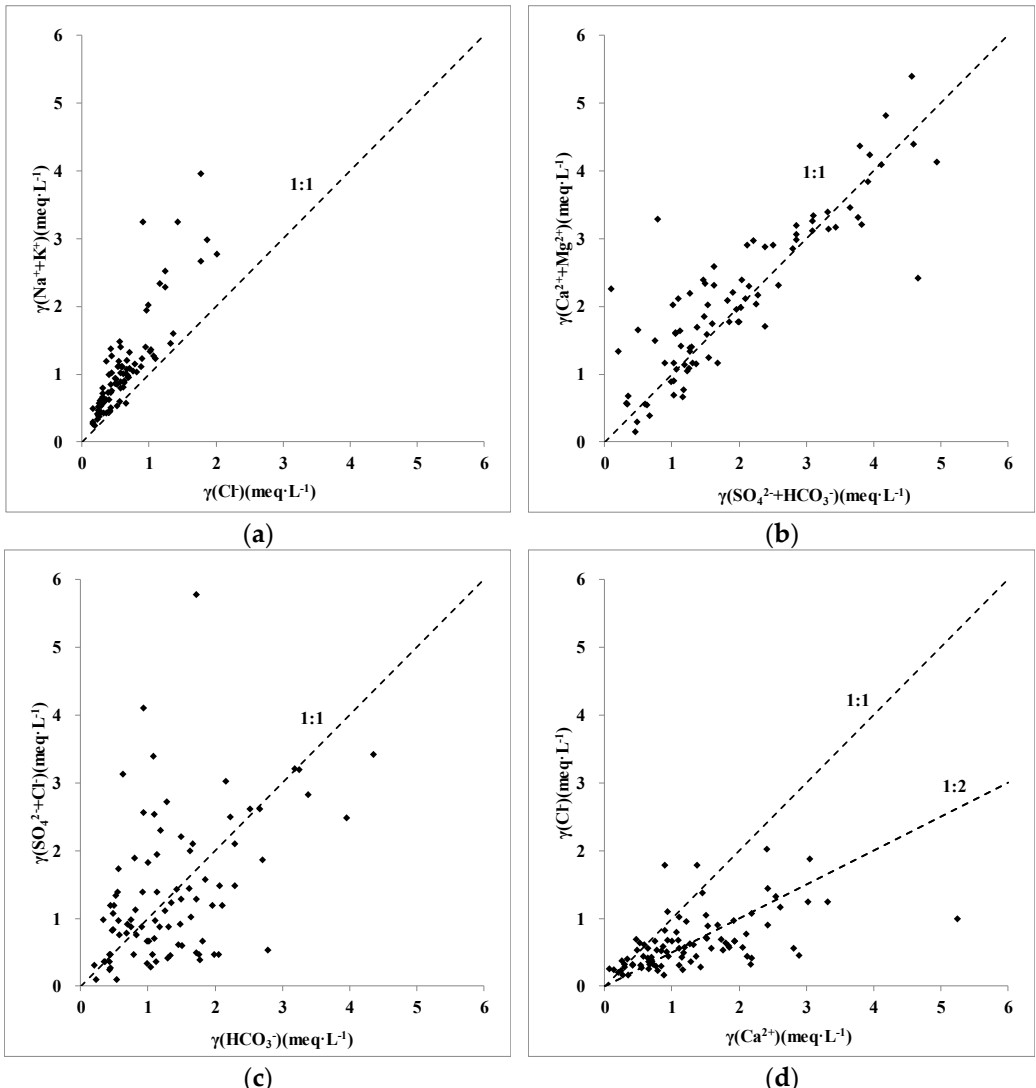

**Figure 6.** Relationship between (**a**) $Na^+ + K^+$ and $Cl^-$; (**b**) $Ca^{2+} + Mg^{2+}$ and $HCO_3^- + SO_4^{2-}$; (**c**) $SO_4^{2-} + Cl^-$ and $HCO_3^-$; (**d**) $Ca^{2+}$ and $Mg^{2+}$.

### 3.3.3. Ion Exchange

The Chlor Alkaline Indices (CAI) are applied to characterize the intensity of ion exchange during the chemical evolution of groundwater [23]. CAI-I and CAI-II are negative when $Na^+$ or $K^+$ exchange $Ca^{2+}$ or $Mg^{2+}$ in the groundwater. However, CAI is positive if reverse ion exchange occurs. As shown in Figure 7, most of the groundwater samples showed negative values for CAI, indicating $Ca^{2+}$ and $Na^+$ had an ion exchange effect in groundwater. In addition, the ratio of $(Na^+ + K^+ - Cl^-)/(Ca^{2+} + Mg^{2+} - HCO_3^- + SO_4^{2-})$ has been widely used to analyze the possible role of ion exchange in groundwater. Moreover, the water samples have a better correlation following the line, indicating that cation exchange has an important influence on groundwater hydrochemistry in the study area.

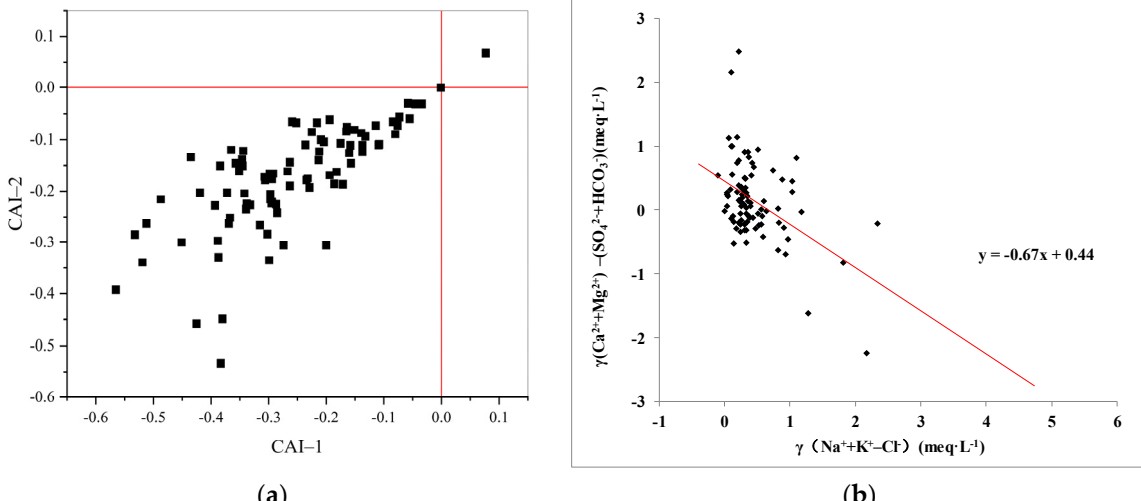

(**a**)　　　　　　　　　　　(**b**)

**Figure 7.** Plots showing (**a**) CAI-I and CAI-II; (**b**) ($Na^+ + K^+ - Cl^-$): ($Ca^{2+} + Mg^{2+} - HCO_3^- + SO_4^{2-}$).

### 3.4. Inverse Geochemical Modeling

PHREEQC software can quantitatively simulate water–rock interactions under the influence of various natural processes and anthropogenic activities [24]. In the current study, two flow paths were selected for simulation along the groundwater flow direction, path 1 (NCX3068→NCX3037) and path 2 (NCX1033→NCX4018).

The "possible mineral phases" are the basis and key to determine the possible reactions along the flow path. Considering the lithology of the strata, diagenetic mineral characteristics of aquifer media, and the hydrochemical analysis results in the study area, calcite, dolomite, sylvite, gypsum, and halite were selected as the main mineral phases for the inverse hydrogeochemical simulation in this work. Since the groundwater system is in an open state, $CO_2(g)$ is regarded as a "possible mineral phase".

The saturation index of main minerals of in groundwater were calculated using PHREEQC. The results of the saturation indices are shown in Table 2. In particular, the value of $SI_{CO2(g)}$ was calculated by $CO_2$ partial pressure, and the formula is $SI_{CO2(g)} = \log10(P_{CO2})$. $P_{CO2}$ is the partial pressure (atm) calculated using activities in the aqueous phase [16]. It can be seen from Table 2, the saturation indices of dolomite, calcite, sylvite, gypsum, and halite were negative, indicating that all these minerals have not reached the saturation state. The unsaturated degree of calcite and dolomite along flow path 1 is obviously alleviated. Halite and sylvite along flow path 2 also show a similar change trend. Thus, the cation exchange between $Na^+$ and $Ca^{2+}$ occurred during mineral dissolution. Specifically, calcium ion from the calcite and gypsum dissolved into the water and then underwent exchange with $Na^+$ absorbed on the rock's surface. The $Na^+$ from the rock surface dissolved into the water, and calcium ion was adsorbed into the rock surface. This can be concluded from the change of ion concentration along the flow path [25] (Figure 8).

**Table 2.** Saturation indices along the flow path 1 and flow path 2.

| Parameters | Flow Paths 1 | | Flow Paths 2 | |
|---|---|---|---|---|
| | NCX3068 | NCX3037 | NCX1033 | NCX4018 |
| $SI_{gypsum}$ | −2.47 | −2.24 | −2 | −2.39 |
| $SI_{calcite}$ | −2.43 | −0.86 | −1.44 | −1.59 |
| $SI_{dolomite}$ | −4.82 | −2.07 | −2.95 | −3.28 |
| $SI_{halite}$ | −7.38 | −6.88 | −7.51 | −6.97 |
| $SI_{sylvite}$ | −7.62 | −6.68 | −7.77 | −7.56 |
| $SI_{CO2(g)}$ | −1.95 | −1.27 | −1.99 | −1.57 |

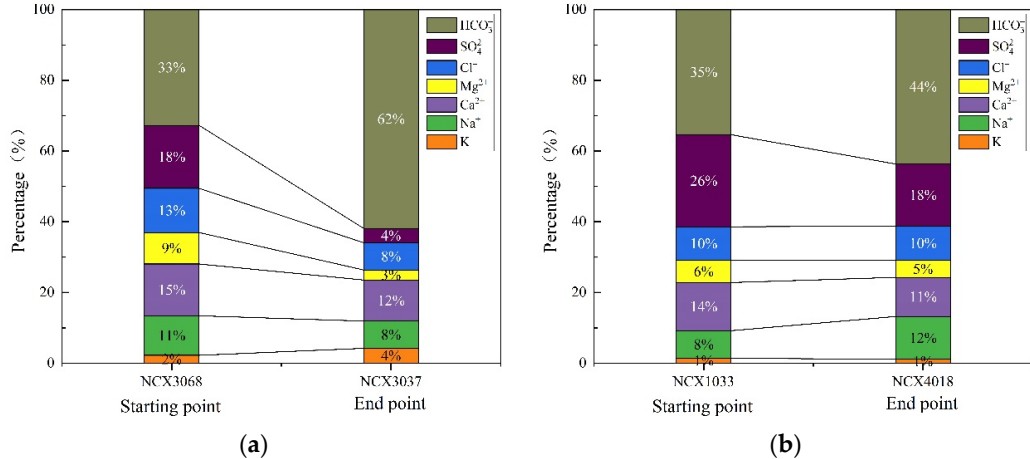

**Figure 8.** Percentage change of ion concentration along the flow path: (**a**) path 1; (**b**) path 2.

The mole transfer of different minerals along groundwater flow paths was calculated by the inverse modeling, and the uncertainty coefficient of model was set to 0.05. The results are summarized in Table 3. The amount of the dissolved gypsum, calcite, dolomite, halite, and sylvite mineral phases along flow path 1 is 0.1157, 0.8789, 0.4096, 1.571, 0.5755, and 3.709 mmol/L, respectively. Moreover, the amount of the calcite, dolomite, and halite mineral that dissolved along flow path 2 is 0.11, 0.029, and 0.9835 mmol/L, respectively, and precipitated gypsum is 0.053 mmol/L.

**Table 3.** The calculation results of mass balance (units: mmol/L).

| Mineral Phase | Flow Paths 1 | Flow Paths 2 |
|---|---|---|
| | NCX3068→NCX3037 | NCX1033→NCX4018 |
| Gypsum | 0.1157 | −0.05306 |
| Calcite | 0.8789 | 0.1106 |
| Dolomite | 0.4096 | 0.02885 |
| Halite | 1.571 | 0.9835 |
| Sylvite | 0.5755 | — |
| $CO_2(g)$ | 3.709 | 0.8482 |

A positive value indicates mineral dissolution, and a negative value indicates mineral precipitation.

The dissolution of $CO_2$ decreased the pH of water, promoting the dissolution of carbonates [26]. As a result of calcite dissolution and ion exchange, the calcium content in the water increases, which in turn leads to gypsum precipitation. It can be concluded that the cation exchange reactions that took place along flow path 2 indicated the dissolution of calcite, halite, and dolomite as the main reason for the precipitation of gypsum.

## 4. Conclusions

This work adopts the methods of mathematical statistics, Piper diagrams, Gibbs diagrams, ion ratios, and hydrogeochemical simulations to study the geochemical characteristics and the controlling factors of the chemical composition of groundwater in a part of the Nanchang section of the Ganfu plain. The results are as follows.

The shallow groundwater is weakly acidic, and $Na^+$ and $Ca^{2+}$ are the most dominant ions. The average concentration of cation in groundwater decreases in $Ca^{2+} > Na^+ > Mg^{2+} > K^+$. $HCO_3^-$ is the most dominant anion, and that anionic concentration is in $HCO_3^- > NO_3^- > SO_4^{2-} > Cl^-$. The coefficients of variation show that $Na^+$ and $SO_4^{2-}$ have significant variability and dispersion in space. The hydrochemical types of groundwater are mainly $HCO_3$-Na·Ca·Mg, $HCO_3$·Cl-Na·Ca·Mg, and $HCO_3$-Na·Ca.

The main controlling factors of groundwater hydrochemistry are water–rock interactions. $Na^+$ and $K^+$ mainly originate from the dissolution of halite. $Ca^{2+}$ and $Mg^{2+}$ are mainly

controlled by carbonate dissolution, while the main anions come from the dissolution of evaporite and carbonate.

According to inverse geochemical modeling results, the evolution of groundwater is affected by the dissolution and precipitation of the mineral phase and cation exchange. And the saturation indices of dolomite, calcite, anhydrite, gypsum, and halite are negative, indicating that these minerals in groundwater are unsaturated. The unsaturated degree of calcite and dolomite along flow path 1 is alleviated. Halite and sylvite along flow path 2 also show a similar change trend. Thus, the cation exchange between $Na^+$ and $Ca^{2+}$ occurred along the path. Moreover, the $Na^+$ from the rock surface dissolved into the water, and the calcium ion was adsorbed into the rock surface.

**Author Contributions:** Methodology, Q.M.; software, Q.M.; writing—original draft preparation, Q.M.; project administration, W.G.; formal analysis, F.T. All authors have read and agreed to the published version of the manuscript.

**Funding:** The research work described herein was funded by a multi-element urban geological survey project of Nanchang city, Nanjing Center, China Geological survey (No: DD20189240).

**Institutional Review Board Statement:** Not applicable.

**Informed Consent Statement:** Written informed consent has been obtained from the patient(s) to publish this paper.

**Data Availability Statement:** Not applicable.

**Acknowledgments:** We appreciate the support from Changjian Chen of School of Water Resources & Environmental Engineering, East China University of Technology. We also thank two anonymous reviewers for their helpful comments and suggestions which have led to significant improvement of this paper.

**Conflicts of Interest:** The authors declare no conflict of interest.

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
