# Peer review of "Geochemical Characteristics and Controlling Factors of Chemical Composition of Groundwater in a Part of the Nanchang Section of Ganfu Plain"

_sustainability, doi:10.3390/su14137976_

Round 1

Reviewer 1 Report

some suggestions:

(1)it's better to give some more describtions on the  analytical method in the section 2.3 including Piper 's diagram,Gibgs's diagram,or else ,there is only one formula in the paper

(2)in the line 72 ,the figure 1 not clear to read 

(3)in the line 150,HCO3+Cl-Na.Ca.Mg facies, it is different with that in the line 262

(4)in the line 200, Figure 2 should be changed to Figure 6

(5)in the line 212 ,Figure 2 should be changed to Figure 7

(6)in the 227, of gypsum should be deleted

(7)in the 245, how to get the number 0.39 mmol/l?

(8)in the 253, piper should be modified as Piper

(9)give more explanation for the figure 5, why  belong to rock-water interaction situations occuring more

Reviewer 2 Report

Manuscript #1766896 presents new data on the groundwater chemistry of the Nanchang region. This is mainly due to the fact that there are some conceptual errors (e.g., the misinterpretation of the CO2 saturation index instead of logPCO2, see comment line 228 below) but also in the structure of the work (there are three Figure 2!). Furthermore: i) some methods of analysis and data interpretation are used without a proper citation of the original works; ii) X-ray analyses of rock/minerals are mentioned in the text, but neither the results are present nor the method in section 2.3 Analytical methods was described. I warmly suggest to upload these results as supplentary files.
The detailed comments to the manuscript are listed below:

line 31: maybe the word "affected" (without bracketing) is better than "branded".

Figure 1: in the China map, on the up left corner, it would be useful to highlight better (by red area or placeholder) the studied area

line 87-88: please specify if water were 0.45 microm filetered and if aliquots for cations acidified to prevent mineral precipitation

line 97: TH = Total Hardness? Please specify the meaning of that acronym. HCO3-: please use the correct subscript (3) and uppescript (-). Moreover, for TH, HCO3, and all other analytical methods used, the brand-model of the instruments and the proper Standards Methods coding should be used: take a look to https://www.standardmethods.org/ or APHA-AWWA-WEF (2017)

line 101: was fluorine analyzed by Ion chromatography or other method (e.g. selective electrode)? Please specify.

line 108: delete And: "The hydrogechemical calculation (saturation indices, logPCO2) and modeling (inverse modeling) were carried out..."

line 109: the mention of Phreeqc necessitates of a proper bibliographic citation: https://pubs.er.usgs.gov/publication/tm6A43. Moreover, please specify the version of the code and the thermodynamic database used for calculations.

line 129-131: the term "young" is improperly used here, because 14C or tritium data are not avaiable to confirm a datation of the water samples. Indeed, most of shallow groundwater of meteoric water has a Ca-HCO3 composition (e.g., Appelo and Postma 2007). Therefore, a discussion in term of depth (shallow or deep) should be more correct.

Table 1: please use the proper upperscript/subscript of ionic species in the parameters column

section 3.3.2 and Figure 2: what is the meaning and utility of gamma letter (γ) there? I am wondering if it would possible to delete it. In water thermodynamics, γ letter is used to indicate the coefficient of activity of a dissolved species. Therefore, to avoid confusion, I suggest to delet it. Please note that the order of figures is completely wrong. Moreover, in figure 2, only letters a) ad b) are shown.

line 158: "which is based on a large number of hydrochemical statistics of natural water bod-157 ies around the world [13]." I don't think so. It should be noted that Gibbs diagrams were orinally drawn using SURFACE water samples. I warmly suggest to check (and to cite in the references) the original work of Gibbs (1970), the recent review of Marandi and Shand (2018) and, after that, rework that section.

line 203: "The present study used the chloralkaline indices (CAI; Schoeller, 1967) to..."; please correct re-numbering the reference citations.

line 217: in Fig.1, please draw the paths by two arrows.

line 220: I will be very curious to observe the results of X-Ray diffraction. Therefore, I suggest to upload the results of X-ray diffraction as supplementary file, along with a quantitative interpretation of them.

line 228: It should be noted that SI-CO2 is not a saturation index, but is the log(10) of CO2 partial pressure. For example, rainwater in equilibrium with atmosphere has a logPCO2 = -3.4 taking into account the present-day concentration of approx. 420 ppm of CO2. Obviously, groundwater has higher logPCO2 values due to bacterial and roots respiration in soil (e.g., Appelo and Postma, 2007). I warmly suggest to re-phrase the section taking into account this informations.

line 247: the solubility of gypsum does not depends by pH: CaSO4(H2O) = Ca2+ + SO4-2 + H2O. On the contrary, carbonates are influnced by pH.

Table 3: please specify the method to calculate the mole transfer. It was calculated by inverse modeling? If yes, please specify the condition  of the modeling (see comment to line 108).

References

APHA-AWWA-WEF (2017) Standard Methods for examination of water and wastewater. Ed.: Baird R.B., Eaton A.D., Rice E.W. 23rd Edition. American Public Health Association (APHA), the American Water Works Association (AWWA), and the Water Environment Federation (WEF).

Appelo, C.A.J., Postma, D. (2005) Geochemistry, Groundwater and Pollution. Second Edition. CRC Press, Taylor & Francis Group, A.A. Balkema Publishers.

Gibbs, R. J. (1970). Mechanisms controlling world water chemistry. Science, 170(3962), 1088-1090.
https://doi.org/10.1126/science.170.3962.1088

Marandi, A., & Shand, P. (2018). Groundwater chemistry and the Gibbs Diagram. Applied Geochemistry, 97, 209-212.
https://doi.org/10.1016/j.apgeochem.2018.07.009

Schoeller, H. (1967). Qualitative evaluation of groundwater resources. In: Methods and techniques of groundwater investigation and development. Water Resources Series No. 33, Unite Nation, New York, , pp.: 44-52

Hope this helps

Round 2

Reviewer 2 Report

The authors have adequately addressed your comments raised in the previous round of review.
However, another small correction and one question (line 336-339 - section 4 Discussion and Conclusions section) occurred to me in reading the revised manuscript:

Figure 1 - Point 1: in the upper China's map, the placeholder is lacking
line 107-Point 3: "0.45 micrometer filtered" or "water was filetered by 0.45 μm pore size membrane"
line 140: AqQA software reference is lacking
line 336-339: if the Na and Ca are from both dissolution of halite-carbonate and clay exchange, I was wondering what's is the budget between the two processes. This should be calculated and specified to avoid confusion in the reader.

Hope this helps
